

# Why choose Random Forest to predict rare species distribution with few samples in large undersampled areas? Three Asian crane species models provide supporting evidence

Chunrong Mi[1], Falk Huettmann[2], Yumin Guo[1], Xuesong Han[1] and Lijia Wen[1]

[1] College of Nature Conservation, Beijing Forestry University, Beijing, China
[2] EWHALE Lab, Department of Biology and Wildlife, Institute of Arctic Biology, University of Alaska Fairbanks (UAF), Fairbanks, AK, United States

Corresponding author
Yumin Guo, guoyumin@bjfu.edu.cn

## ABSTRACT

Species distribution models (SDMs) have become an essential tool in ecology, biogeography, evolution and, more recently, in conservation biology. How to generalize species distributions in large undersampled areas, especially with few samples, is a fundamental issue of SDMs. In order to explore this issue, we used the best available presence records for the Hooded Crane (*Grus monacha*, $n = 33$), White-naped Crane (*Grus vipio*, $n = 40$), and Black-necked Crane (*Grus nigricollis*, $n = 75$) in China as three case studies, employing four powerful and commonly used machine learning algorithms to map the breeding distributions of the three species: TreeNet (Stochastic Gradient Boosting, Boosted Regression Tree Model), Random Forest, CART (Classification and Regression Tree) and Maxent (Maximum Entropy Models). In addition, we developed an ensemble forecast by averaging predicted probability of the above four models results. Commonly used model performance metrics (Area under ROC (AUC) and true skill statistic (TSS)) were employed to evaluate model accuracy. The latest satellite tracking data and compiled literature data were used as two independent testing datasets to confront model predictions. We found Random Forest demonstrated the best performance for the most assessment method, provided a better model fit to the testing data, and achieved better species range maps for each crane species in undersampled areas. Random Forest has been generally available for more than 20 years and has been known to perform extremely well in ecological predictions. However, while increasingly on the rise, its potential is still widely underused in conservation, (spatial) ecological applications and for inference. Our results show that it informs ecological and biogeographical theories as well as being suitable for conservation applications, specifically when the study area is undersampled. This method helps to save model-selection time and effort, and allows robust and rapid assessments and decisions for efficient conservation.

## INTRODUCTION

Species distribution models (SDMs) are empirical ecological models that relate species observations to environmental predictors (*Guisan & Zimmermann, 2000*; *Drew, Wiersma & Huettmann, 2011*). SDMs have become an increasingly important and now essential tool in ecology, biogeography, evolution and, more recently, in conservation biology (*Guisan et al., 2013*), management (*Cushman & Huettmann, 2010*), impact assessments (*Humphries & Huettmann, 2014*) and climate change research (*Lei et al., 2011*; *Mi, Huettmann & Guo, 2016*). To generalize and infer from a model, or model transferability is defined as geographical or temporal cross-applicability of models (*Thomas & Bovee, 1993*; *Kleyer, 2002*; *Randin et al., 2006*). It is one important feature in SDMs, a base-requirement in several ecological and conservation biological applications (*Heikkinen, Marmion & Luoto, 2012*). In this study, we used generality (transferability) as the concept of generalizing distribution from sampled areas to unsampled areas (extrapolation beyond the data) in one study area.

Detailed distribution data for rare species in large areas are rarely available in SDMs (*Pearson et al., 2007*; *Booms, Huettmann & Schempf, 2010*). However, they are among the most needed for their conservation to be effective. Collecting and assembling distribution data for species, especially for rare or endangered species in remote wilderness areas is often a very difficult task, requiring a large amount of human, time and funding sources (*Gwena et al., 2010*; *Ohse et al., 2009*).

Recent studies have suggested that machine-learning (ML) methodology, may perform better than the traditional regression-based algorithms (*Elith et al., 2006*). TreeNet (boosting; *Friedman, 2002*), Random Forest (bagging; *Breiman, 2001*), CART (*Breiman et al., 1984*) and Maxent (*Phillips, Dudík & Schapire, 2004*) are considered to be among the most powerful machine learning algorithms and for common usages (*Elith et al., 2006*; *Wisz et al., 2008*; *Williams et al., 2009*; *Lei et al., 2011*) and for obtaining powerful ensemble models (*Araújo & New, 2007*; *Hardy et al., 2011*). Although *Heikkinen, Marmion & Luoto (2012)* compared the four SDMs techniques' transferability in their study, they did not test with rare species and few samples in undersampled areas. It is important to understand that the software platform of the former three algorithms (Boosted Regression Trees, Random Forest and CART) applied by *Heikkinen, Marmion & Luoto (2012)* from the R software ("BIOMOD" framewok) comes without a GUI and lacks sophisticated optimization, sample balancing and fine-tuning, but as they are commonly used though by numerous SDM modelers. Instead, we here run these models in the Salford Predictive Modeler (SPM version 7) by Salford Systems Ltd. (https://www.salford-systems.com/). These algorithms in SPM are further optimized and improved by one of the algorithm's original co-authors (especially for TreeNet and Random Forest). It runs with a convenient GUI, and produces a number of descriptive results and graphics which are virtually not available in the R version. While this is a commercial software, it is usually available on a 30 days trial version (which suffices for most model runs we know. Also, some of the features of the randomForest R package, most notably the ability to produce partial dependence plots (*Herrick, 2013*), are not directly implemented yet in SPM7 (but they can essentially be obtained by running TreeNet in a Random Forest model).

Model generality (transferability) testing could offer particularly powerful for model evaluation (*Randin et al., 2006*). Independent observations from a training data set has been recommended as a more proper evaluations of models (*Fielding & Bell, 1997*; *Guisan & Zimmermann, 2000*). Therefore, the use of an independent geographically (*Fielding & Haworth, 1995*) or temporally (*Boyce et al., 2002*; *Araújo et al., 2005*) testing data set is encouraged to assess the generality of different SDMs techniques. Data from museum specimen, published literature (*Graham et al., 2004*) as well as tracking are good source to assess model generality (transferability) performance. In addition, how the distribution map links with reality data, especially in undersampled areas where modelers want to make predictions should definitely be employed as a metric to assess model performance and generalization. Arguably, if model predictions perform very well there, great progress is provided and usually done cost-effective too. However, predictions on existing knowledge and data offer less progress. The model prediction and conservation frontier obviously sits in the unknown and to provide progress there (*Breiman, 2001*; *Drew, Wiersma & Huettmann, 2011*).

In this study, we investigated models for the best-available data for three species in East Asia as test cases: Hooded Cranes (*Grus monacha*, $n = 33$), White-naped Cranes (*Grus vipio*, $n = 40$) and Black-necked Cranes (*Grus nigricollis*, $n = 75$). Four machine-learning model algorithms (TreeNet, Random Forest, CART and Maxent) were applied to map breeding distributions for these three crane species which otherwise lack empirically derived distribution information. In addition, two kinds of independent testing data sets (latest satellite tracking data, and compiled literature data (Threatened Birds of Asia: *Collar, Crosby & Crosby, 2001*) were obtained to test the transferability of the four model algorithms. The purpose of this investigation is to explore whether there is a SDM technique among the four algorithms that could generate reliable and accurate distributions with high generality for rare species using few samples but in large undersampled areas. Results from this research could be useful for the detection of rare species and enhance fieldwork sampling in large undersampled areas which would save money and effort, as well as advance the conservation management of those species.

## MATERIALS AND METHODS

### Species data
In our 13 combined years of field work, we have collected 33 Hooded Crane nests (2002–2014), 40 White-naped Crane nests (2009–2014) (Supplemental Information 2), and 75 Black-necked Crane nests (2014) (see Fig. 1), during breeding seasons. We used these field samples (nests) to represent species presence points referenced in time and space.

### Environmental variables
We used 21 environmental layers at a 30-s spatial resolution in a GIS format and that were known to correlate with bird distribution and as proxies of habitats predictors. They included bio-climatic factors (bio_1-7, bio_12-15), topographical factors (altitude, slope, and aspect), water factors (distance to river, distance to lake, and distance to coastline), inference factors (distance to road, distance to rail road, and distance to settlements), and

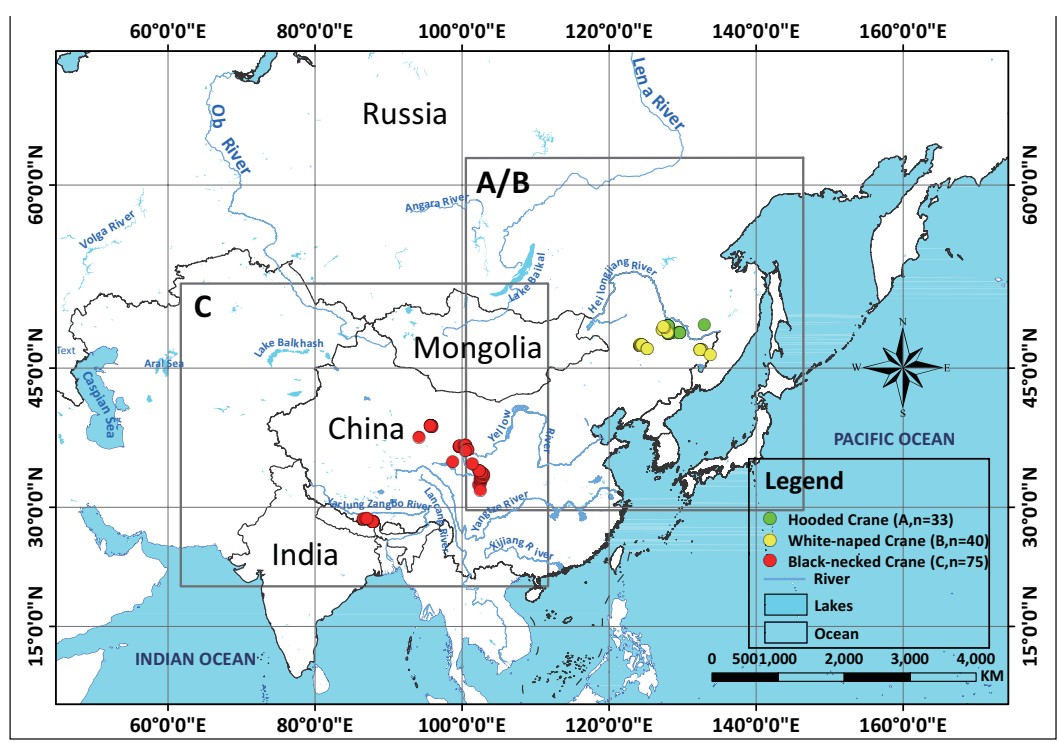

**Figure 1** Study areas for three species cranes.

land cover factors (for detailed information, see Table 1). Most of these predictors were obtained from open access sources. Bio-climate factors were obtained from the WorldClim database (http://www.worldclim.org), while aspect and slope layer were derived from the altitude layer in ArcGIS, which was also initially obtained from the WorldClim database. Road, railroad, river, lake and coastline and settlement maps were obtained from the Natural Earth database (http://www.naturalearthdata.com). The land cover map was obtained from the ESA database (http://www.esa-landcover-cci.org). We also made models with all 19 bio-climate variables and 10 other environmental variables, and then reduced predictors by AIC, BIC, varclust, PCA and FA analysis. When we compared the distribution maps overlaying with independent data set generated by Random Forest model, we found the model based on 21 predictors have the best performance for Hooded Cranes, and the best level for White-naped Crane and Black-necked Cranes (see Table S1). Therefore, we decided to constructed models with 21 predictors for the all three cranes and four machine-learning techniques. All spatial layers of these environmental variables were resampled in ArcGIS to a resolution of 30-s to correspond to that of the bioclimatic variables and for a meaningful high-resolution management scale.

## Model development

We created TreeNet, Random Forest, CART, Maxent models and Ensemble models (averaged value of the former four model results) for Hooded Cranes, White-naped Cranes and Black-naped Cranes. These four model algorithms are considered to be among the best

**Table 1  Environmental GIS layers used to predict breeding distributions of three cranes.**

| Environmental layers | Description | Source | Website |
|---|---|---|---|
| Bio_1 | Annual mean temperature (°C) | WorldClim | http://www.worldclim.org/ |
| Bio_2 | Monthly mean (max temp–min temp) (°C) | WorldClim | http://www.worldclim.org/ |
| Bio_3 | Isothermality (BIO2/BIO7) (*100 °C) | WorldClim | http://www.worldclim.org/ |
| Bio_4 | Temperature seasonality (standard deviation *100 °C) | WorldClim | http://www.worldclim.org/ |
| Bio_5 | Max temperature of warmest month (°C) | WorldClim | http://www.worldclim.org/ |
| Bio_6 | Min temperature of coldest month (°C) | WorldClim | http://www.worldclim.org/ |
| Bio_7 | Annual temperature range (BIO5-BIO6) (°C) | WorldClim | http://www.worldclim.org/ |
| Bio_12 | Annual precipitation (mm) | WorldClim | http://www.worldclim.org/ |
| Bio_13 | Precipitation of wettest month (mm) | WorldClim | http://www.worldclim.org/ |
| Bio_14 | Precipitation of driest month (mm) | WorldClim | http://www.worldclim.org/ |
| Bio_15 | Precipitation seasonality (mm) | WorldClim | http://www.worldclim.org/ |
| Altitude | Altitude (m) | WorldClim | http://www.worldclim.org/ |
| Aspect | Aspect (°) | Derived from Altitude | http://www.worldclim.org/ |
| Slope | Slope | Derived from Altitude | http://www.worldclim.org/ |
| Landcover | Land cover | ESA | http://www.esa-landcover-cci.org/ |
| Disroad | Distance to roads (m) | Road layer from Natural Earth | http://www.naturalearthdata.com/ |
| Disrard | Distance to railways (m) | Railroad layer from Natural Earth | http://www.naturalearthdata.com/ |
| Disriver | Distance to rivers (m) | River layer from Natural Earth | http://www.naturalearthdata.com/ |
| Dislake | Distance to lakes (m) | Lake layer from Natural Earth | http://www.naturalearthdata.com/ |
| Discoastline | Distance to coastline (m) | Coastline layer from Natural Earth | http://www.naturalearthdata.com/ |
| Dissettle | Distance to settlements (m) | Settle layer from Natural Earth | http://www.naturalearthdata.com/ |

performing machine learning methods (more information about these four models can be seen in the references by *Breiman et al., 1984*; *Breiman, 2001*; *Friedman, 2002*; *Phillips, Dudík & Schapire, 2004*; *Hegel et al., 2010*). The first three machine learning models are binary (presence-pseudo absence) models and were handled in Salford Predictive Modeler 7.0 (SPM). For more details on TreeNet, Random Forest and CART in SPM and their performances, we refer readers to the user guide document online (https://www.salford-systems.com/products/spm/userguide). Several implementations of these algorithms exist. Approximately 10,000 'pseudo-absence' locations were selected by random sampling across the study area for each species using the freely available Geospatial Modeling Environment (GME; Hawth's Tools; *Beyer, 2013*; see *Booms, Huettmann & Schempf, 2010* and *Ohse et al., 2009* for examples). We extracted the habitat information from the environmental layers for presence and pseudo-absence points for each crane, and then constructed models in SPM with these data. In addition, we used balanced class weights, and 1,000 trees were built for all models to find an optimum within, others used default settings.

For the predictions, we created a 'lattice' (equally spaced points across the study area; approximately 5 × 5 km spacing for the study area). For the lattice, we extracted information from the same environmental layers (Table 1) as described above for each point and then used the model to predict ('score') bird presence for each of the regular lattice points. For

visualization, we imported the dataset of spatially referenced predictions ('score file') into GIS as a raster file and interpolated for visual purposes between the regular points using inverse distance weighting (IDW) to obtain a smoothed predictive map of all pixels for the breeding distributions of the three cranes (as performed in *Ohse et al., 2009*; *Booms, Huettmann & Schempf, 2010*). The fourth algorithm we employed, Maxent, is commonly referred to as a presence-only model; we used Maxent 3.3.3 k (it can be downloaded for free from http://www.cs.princeton.edu/~schapire/maxent/) to construct our models. To run Maxent, we followed the 3.3.3e tutorial for ArcGIS 10 (*Young, Carter & Evangelista, 2011*) and used default settings.

## Testing data and model assessment

We applied two types of testing data in this study: one consisted of satellite tracking data, and the other was represented by data from the literature. Satellite tracking data were obtained from four individual Hooded Cranes and eight White-naped Cranes that were tracked at stopover sites (for more details regarding the information for tracked cranes, please see Fig. S1). The satellite tracking devices could provide 24 data points per day (Supplemental Information 4). Here, we chose points that had a speed of less than 5 km/h during the period from 1st May to 31th June for Hooded Cranes and 15th April to 15th June for White-naped Cranes as the locations of the breeding grounds for these two cranes. The total numbers of tracking data points were 4,963 and 7,712 (Hooded Cranes and White-naped Crane, respectively. We didn't track Black-necked Cranes, so there was no tracking testing data for this species.). The literature data for this study were obtained by geo-referencing the location points of detections from 1980–2000 (ArcGIS 10.1) from Threatened Birds of Asia: the BirdLife International Red Data Book (*Collar, Crosby & Crosby, 2001*). From this hardcopy data source, we were able to obtain and digitize 27 breeding records for Hooded Cranes, 43 breeding records for White-naped Cranes, and 53 breeding records for Black-necked Cranes (see Figs. 2A–2C and Supplemental Information 3). Here we digitized the only available crane data for these three species in East-Asia into a database.

In addition, we generated 3,000 random points for Hooded Cranes and White-naped Cranes, and 5,000 random points for Black-necked Cranes as testing pseudo-absence points in their respective study areas; then, the literature locations (additional presence points for testing) and random points locations (testing absence points) that contrasted with the associated predictive value of RIO extracted from the relative prediction map, which were used to calculate receiver operating characteristic (ROC) curves and the true skill statistic (TSS) (*Hijmans & Graham, 2006*). The area under the ROC curve (AUC) is commonly used to evaluate models in species distributional modeling (*Manel, Williams & Ormerod, 2001*; *McPherson, Jetz & Rogers, 2004*). TSS was also used to evaluate model performance; we used TSS because it has been increasingly applied as a simple but robust and intuitive measure of the performance of species distribution models (*Allouche, Tsoar & Kadmon, 2006*).

To assess models transferability, we extracted the predictive value of the relative index of occurrence (RIO) for testing data sets from the prediction maps using GME. We then constructed resulting violin plots in R for these extracted RIOs to visualize their one-dimensional distribution. This method allowed us to examine the degree of generalizability

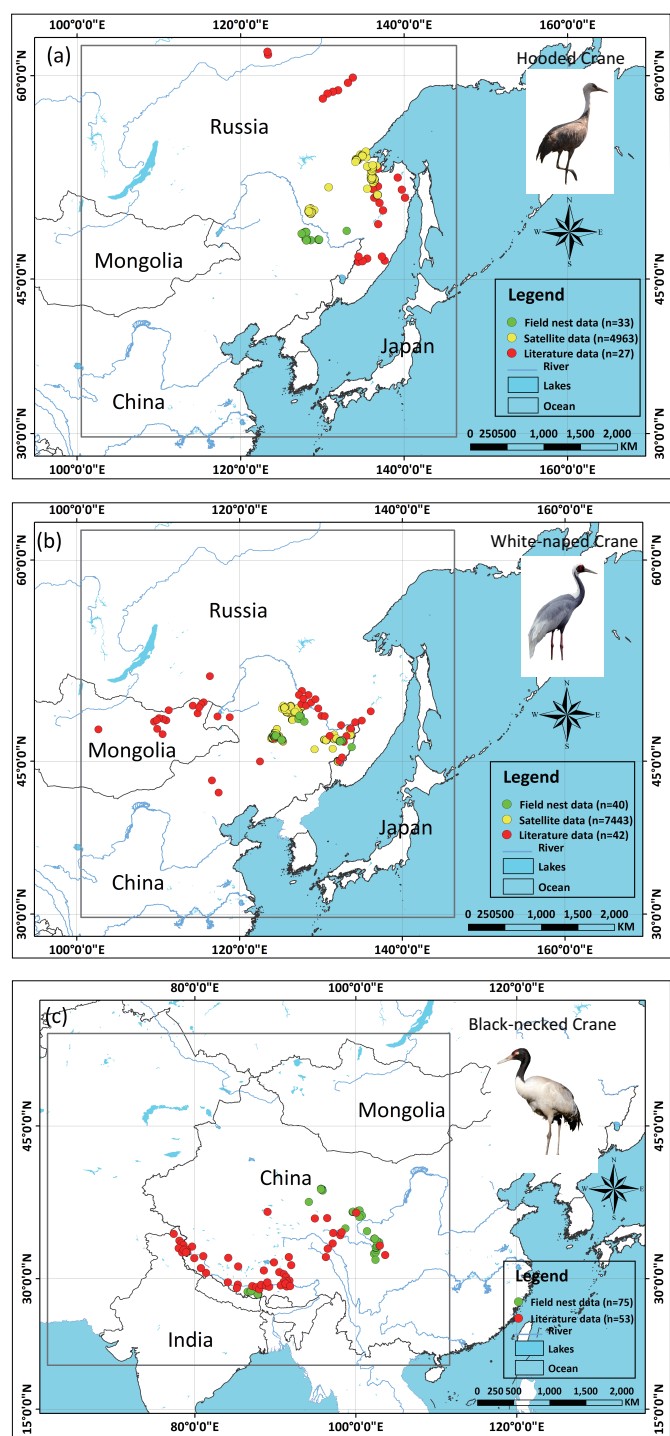

**Figure 2  Detailed study areas showing the presence of and testing data used for the three cranes.** (A) Hooded Cranes, (B) White-naped Cranes, (C) Black-necked Cranes.

**Table 2** AUC and TSS values for four machine learning models and their Ensemble model with three crane species based on literature testing data.

| Accuracy metric (samples) | Species distribution model | | | | |
|---|---|---|---|---|---|
| | TreeNet | Random Forest | CART | Maxent | Ensemble |
| Hooded Crane (*Grus monacha, n* = 33 sites) | | | | | |
| AUC | 0.504 | **0.625** | 0.500 | 0.558 | 0.558 |
| TSS | 0.000 | **0.250** | 0.000 | 0.137 | 0.117 |
| White-naped Crane (*Grus vipio, n* = 40 sites) | | | | | |
| AUC | 0.605 | **0.754** | 0.564 | 0.712 | **0.754** |
| TSS | 0.210 | **0.509** | 0.128 | 0.424 | 0.508 |
| Black-necked Crane (*Grus nigricollis, n* = 75 sites) | | | | | |
| AUC | 0.528 | 0.830 | 0.672 | 0.805 | **0.843** |
| TSS | 0.055 | 0.660 | 0.345 | 0.611 | **0.686** |

based on the local area with samples to predict into undersampled areas that are otherwise unsampled in the model development (=areas without training data). In addition, AUC is also commonly used to assess model transferability in our study referring *Randin et al. (2006)*.

## RESULTS

### Model performance

The results for AUC and TSS, two metrics commonly used to evaluate model accuracy, are listed in Table 2. For the four SDMs technique, our results showed that the AUC values for Random Forest were always highest (>0.625), ranking this model in first place, followed by Maxent (>0.558), and then either CART or TreeNet (≥0.500). TSS showed us consistent results, as was the case for AUC, and Random Forest performed the best (>0.250) followed by Maxent (>0.137) for all three crane species, CART took the third place for Black-necked Cranes, and TreeNet performed better than CART for White-naped Cranes. And the results showed there was a trend that the value of these three metrics increased with an increase of nest site samples (33–75, Hooded Crane to Black-necked Crane, see Table 2). Comparing the results of Random Forest with Ensemble models, we found their performance were close. Random Forest obtained better models for Hooded Cranes and White-naped Cranes cases, the Ensemble model performed better for Black-necked Cranes.

### Model generalization

Violin plots for RIOs with overlaid satellite tracking data (Fig. 3) showed that Random Forest for Hooded Cranes and White-naped Cranes performed better than the other three models. In the Hooded Crane models (Fig. 3A), the RIO for most satellite tracking data indicated that TreeNet, and CART predicted with a value around 0; Ensemble model demonstrated a slightly higher value than the other three models but was still much lower than Random Forest. Figure 3B indicates the same situation than found in Fig. 3A: Random Forest still performed better than the other three models (median values in Random Forests were close to 1.00). TreeNet had a median RIO value of approximately 0.71, followed by

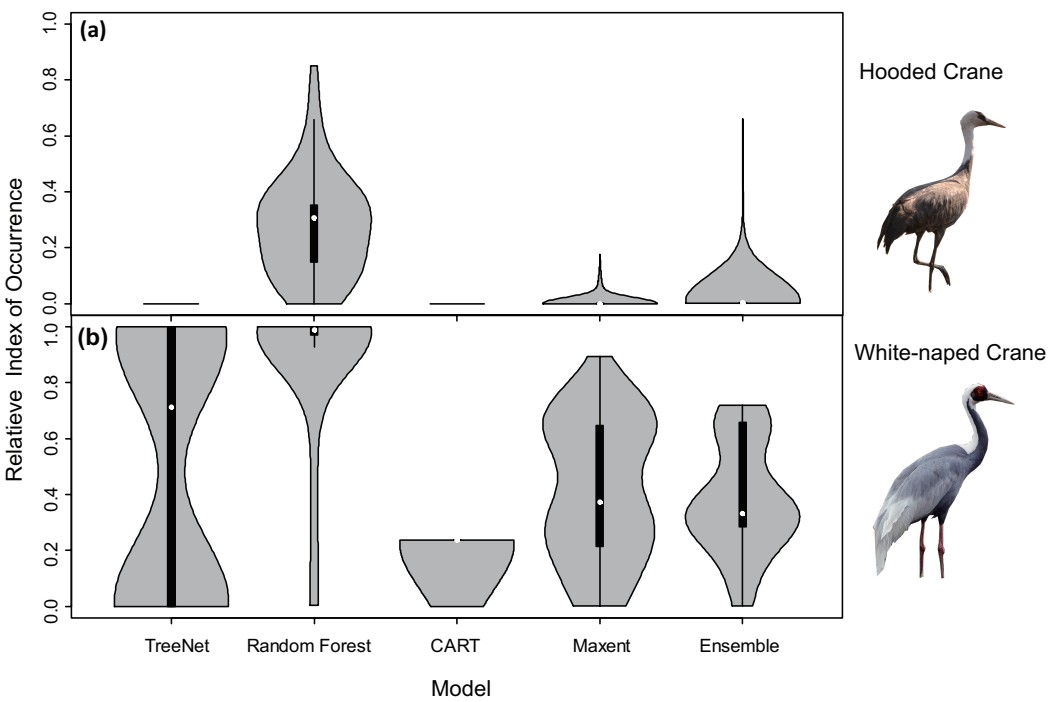

**Figure 3** **Violin plots of the Relative Index of Occurrence (RIO) for four SDMs and Ensemble model for Hooded Cranes and White-naped Cranes based on satellite tracking data.** (A) Violin plots of Hooded Cranes, (B) violin plots of White-naped Cranes.

Maxent (median was 0.37) and then Ensemble and CART. While some tracking points had a low RIO value in TreeNet, the majority of RIO values for CART remained in the 0.20 range.

Violin plots of the RIOs values for the three cranes extracted for the literature data from the prediction maps (Fig. 4) demonstrated consistent trends (Fig. 3), indicating that Random Forest performed best across all models of the three species. In Fig. 4A, the RIO values for Random Forest ranged from 0 to 0.48, and most RIO values were below 0.1; the RIO values for the other three SDMs method were 0, the Ensemble model performed a little bit better. As showed in Fig. 4B, most RIO values for Random Forest were below 0.7, and the median value was approximately 0.20, followed by Maxent and then CART. The violin plots for Black-necked Cranes (Fig. 4C) indicated that TreeNet performed the worst, although there were some pixels that had high RIO values, followed by the Ensemble model and then Maxent. The best performer was still Random Forest, and its RIOs were distributed evenly to a certain extent with a median value of 0.44. The results of AUC, as mentioned in the "Model performance" section (Table 2), showed consistent results with violin plots, and Random Forest always achieved the highest value and had the best generalization.

## Spatial assessment using a testing data overlay prediction map

An assessment of niche prediction beyond the local area where samples were located represents a real test of the generalizability of the model predictions in undersampled areas. This approach was used to evaluate whether testing data (satellite tracking data and literature data) locations matched predictions of the potential distribution area, as a spatial assessment

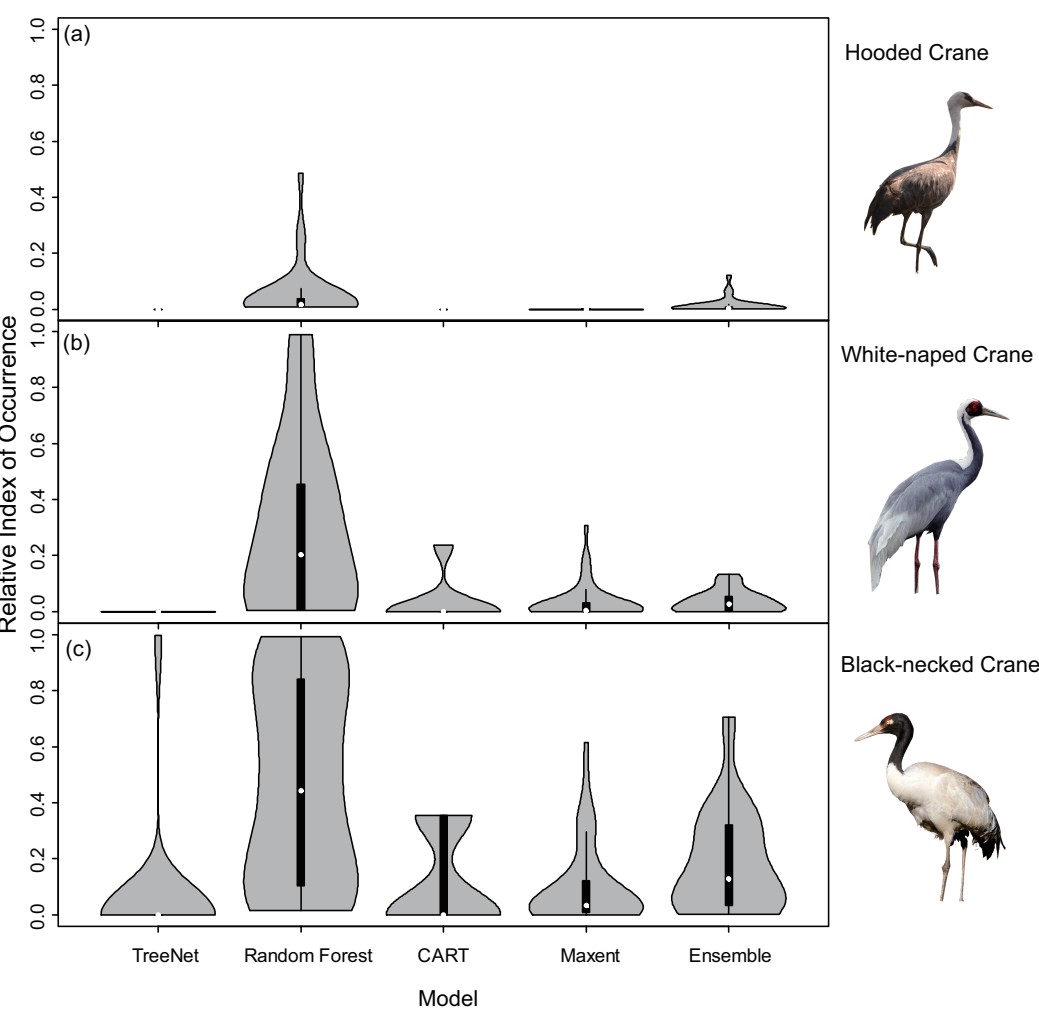

**Figure 4** **Violin plots of Relative Index of Occurrence (RIO) values for four SDMs and Ensemble model for three cranes based on calibration data from Threatened Birds of Asia.** (A) Violin plots for Hooded Cranes, (B) violin plots for White-naped Cranes, (C) violin plots for Black-necked Cranes.

of model performance. It is a spatial and visual method to show the transferability of SDMs from sampled to unsampled areas. From the results (Figs. 5, 6, 7 and Supplemental Information 1), we found that Random Forest demonstrated the strongest performance to handle generality (transferability), and a high fraction of testing data locations were predicted in the distribution areas of the three cranes (Figs. 5B, 5G, 6B, 6G, 7B, 7G). The order of the generality of the remaining four models was: Ensemble model followed by Maxent, CART and then TreeNet. Note, however, that the capacities of these models to predict well in undersampled areas were weaker than Random Forest, it holds particularly for areas that were further away from the sample areas (Figs. 5–7). In addition, we found that the generality increased with sample size (33–75, Hooded Crane to Black-necked Crane, see Figs. 5–7). This means, as expected and known, that a higher sample size makes models more robust and better to generalize from.

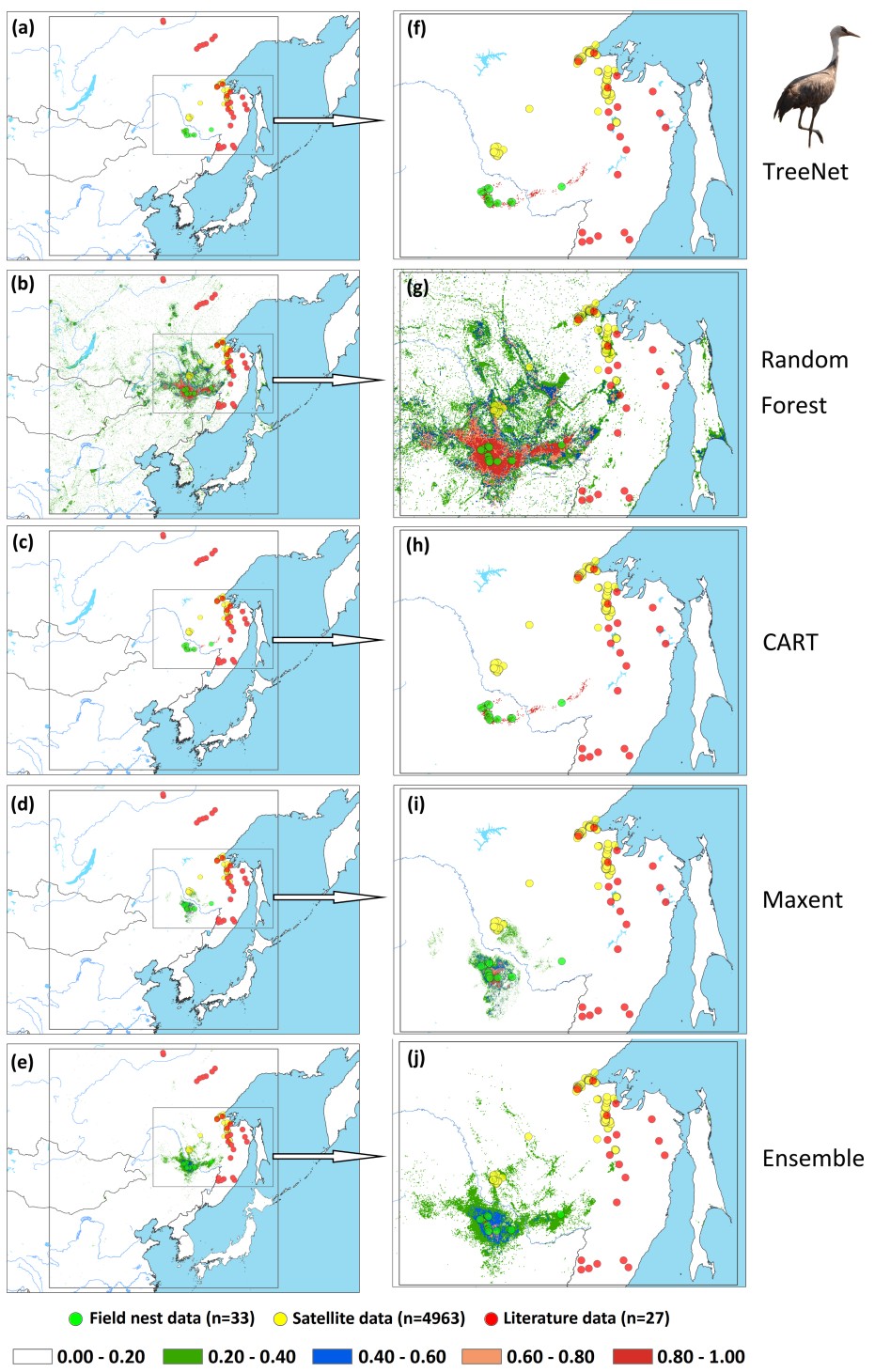

**Field nest data (n=33)** ⬤   **Satellite data (n=4963)** ⬤   **Literature data (n=27)** ⬤

☐ 0.00 - 0.20   ☐ 0.20 - 0.40   ☐ 0.40 - 0.60   ☐ 0.60 - 0.80   ☐ 0.80 - 1.00

**Figure 5** **Prediction maps for Hooded Cranes and zoomed-in maps showing the four models (TreeNet, Random Forest, CART and Maxent) and Ensemble model in detail.** (A–E) Prediction map for Hooded Cranes, (F–J) zoomed-in map for Hooded Cranes.
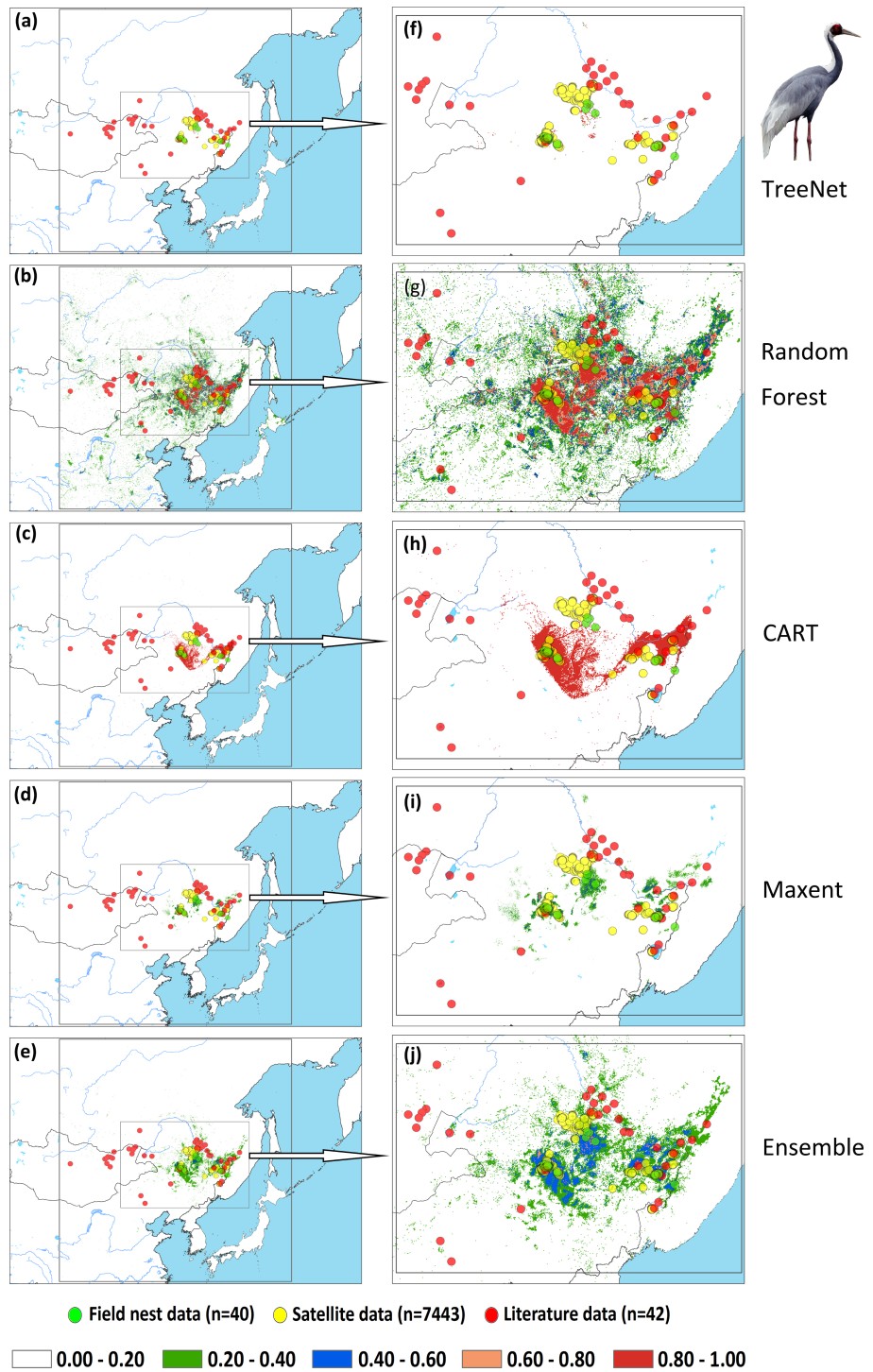

**Field nest data (n=40)** ⬤  **Satellite data (n=7443)** ⬤  **Literature data (n=42)** ⬤

☐ 0.00 - 0.20  ■ 0.20 - 0.40  ■ 0.40 - 0.60  ■ 0.60 - 0.80  ■ 0.80 - 1.00

**Figure 6** **Prediction maps for White-naped Cranes and zoomed-in maps showing the four models (TreeNet, Random Forest, CART and Maxent) and Ensemble model in detail.** (A–E) Prediction map for White-naped Cranes, (F–J) zoomed-in map for White-naped Cranes.

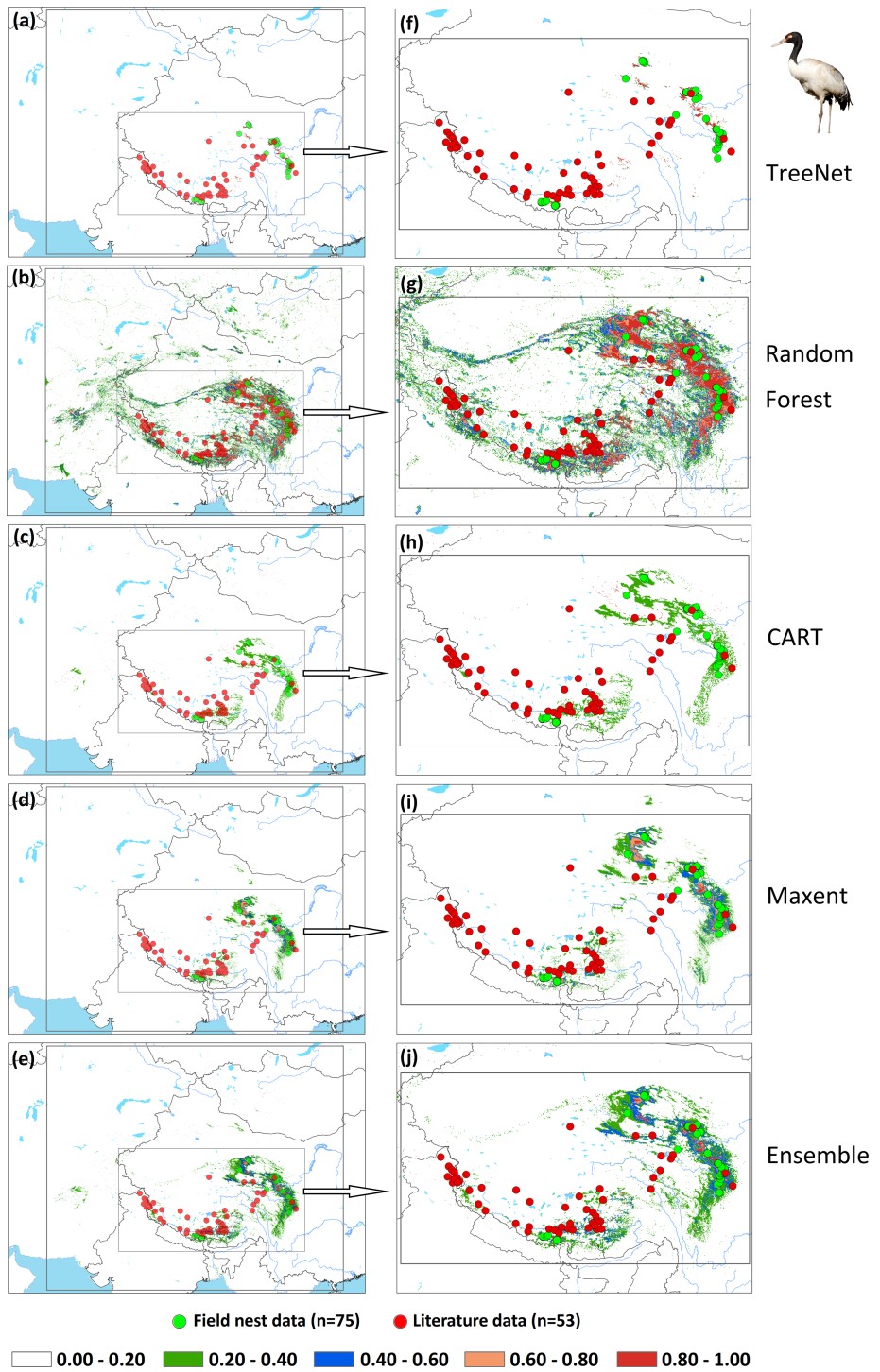

Field nest data (n=75)    Literature data (n=53)

0.00 - 0.20    0.20 - 0.40    0.40 - 0.60    0.60 - 0.80    0.80 - 1.00

**Figure 7** **Prediction maps for Black-necked Cranes and zoomed-in maps showing the four models (TreeNet, Random Forest, CART and Maxent) and Ensemble model in detail.** (A–E) Prediction map for Black-necked Cranes, (F–J) zoomed-in map for Black-necked Cranes.

## DISCUSSION

### Model generality (transferability)

Estimating species distributions in undersampled areas is a fundamental problem in ecology, biogeography, biodiversity conservation and natural resource management (*Drew, Wiersma & Huettmann, 2011*). That is specifically true for rare and difficult to be detected species and which are usually high on the conservation priority. The use of SDMs and with machine learning has become the method for deriving such estimates (*Guisan & Thuiller, 2005*; *Drew, Wiersma & Huettmann, 2011*; *Guisan et al., 2013*) and could contribute to detect new and to confirm populations of rare species. However, the application of a few samples to project a distribution area widely beyond the sample range is a greater challenge and has rarely been attempted in the literature until recently. Only now have conservationists realized its substantial value for pro-active decision making in conservation management (see work by *Ohse et al., 2009*; *Drew, Wiersma & Huettmann, 2011*; *Kandel et al., 2015* etc.). Our results based on AUC, violin plots for RIOs and spatial assessment of testing data (satellite tracking data and literature data) all suggest there are difference in the generalization performance of different modeling techniques (TreeNet, Random Forest, CART and Maxent).

Moreover, among the acknowledged four rather powerful and commonly used machne-learning techniques, Random Forest (bagging) in SPM usually had the best performance in each case. Our results are in agreement with those of *Prasad, Iverson & Liaw (2006)*, *Cutler et al. (2007)* and *Syphard & Franklin (2009)* indicating a superiority of Random Forest in such applications. However, initially it appears to run counter to the conclusions off recent paper (*Heikkinen, Marmion & Luoto, 2012*) with the poor transferability of Random Forest. But we propose this is due to the fact that many Random Forest implementations exist (see the 100 classifier paper *Fernández-Delgado et al., 2014*).

Here we applied Random Forest in SPM which has been optimized under one of the algorithm's original co-authors, while *Heikkinen, Marmion & Luoto (2012)* run a basic Random Forest with BIOMOD framework in the R sofeware and which remains widely un-tuned and largely behind the potential. The differences are known to be rather big (see *Herrick, 2013* for a comparison).

Furthermore, Maxent, a widely used SDM-method consequently greatly enjoyed by many modelers (*Phillips, Anderson & Schapire, 2006*; *Peterson, Monica & Muir, 2007*; *Phillips & Dudík, 2008*; *Li et al., 2015*, and so on), didn't perform so good in regards to transferability in this study. This contrasts to those of *Elith et al. (2006)* and *Heikkinen, Marmion & Luoto (2012)*, where Maxent and GBM (generalized boosting methods) perform well. We infer this may be caused by sample size used as training data and due to the actual algorithms used. When the sample size increased (33–75), the AUC and TSS value of all models rose (Table 2). This indicates that higher sample sizes make models more robust and performing better. Sample sizes of 33 presence points still favor by Random Forest.

In Random Forest, random samples from rows and predictors are used to build hundreds of trees. Each individual tree is constructed from a bootstrap sample and split at each node by

the best predictor from a very small, randomly chosen subset of the predictor variable pool (*Herrick, 2013*). These trees comprising the forest are each grown to maximal depth, and predictions are made by averaged trees through 'voting' (*Breiman, 2001*). This algorithm avoids overfitting by controlling the number of predictors randomly used at each split, using means of out-of-bag (OOB) samples to calculate an unbiased error rate. And also, Random Forest in SPM utilizes additional specific fine-tuning for best performance.

### RIOs of random points

In order to explore whether Random Forest created higher RIOs for prediction maps in each grid, which would result into higher RIOs of testing data, we generated 3,000 random points for Hooded Cranes and White-naped Cranes, 5,000 random points for Black-necked Cranes in their related projected study areas. We made violin plots for RIOs of random points (Fig. 8), and we found that more RIO values of random points for Maxent, Random Forest and Ensemble models were close to the lower value, and then followed by TreeNet. The distribution shapes of Random Forest, Maxent and Ensemble model are more similar to the real distribution of species in the real world. The RIOs of White-naped Crane extracted from the CART model distributed in the range of the low value; this means there were no points located in the high RIO areas of cranes, which is unrealistic. Consequently, we argued that Random Forest did not create higher RIOs for prediction maps in each grid in our study.

### Models with small sample sizes

Conservation biologists are often interested in rare species and seek to improve their conservation. These species typically have limited number of available occurrence records, which poses challenges for the creation of accurate species distribution models when compared with models developed with greater numbers of occurrences (*Stockwell & Peterson, 2002*; *McPherson, Jetz & Rogers, 2004*; *Hernandez et al., 2006*). In this study, we used three crane species as case studies, and their occurrence records (nests) totaled 33, 40, and 75, respectively (considering the small numbers of samples and given that a low fraction of the area was sampled in the large projected area). In our models, we found that model fit (AUC and TSS, see Table 2) of Random Forest that had the highest index, while Maxent usually ranked second. In addition, we found that models with few presence samples can also generate accurate species predictive distributions (Figs. 3–7) with the Random Forest method. Of course, models constructed with few samples underlie the threat of being biased more because few samples usually had not enough information including all distribution gradients conditions of a species, especially for places far away from the location of training presence points. However, the potential distribution area predicted by SDMs could become the place where scholars could look for the birds (additional fieldwork sampling). Also, these places could be used as diffusion or reintroduction areas! It is valuable and new information either way.

### Evaluation methods

In this study, we applied two widely-used assessment methods (AUC and TSS) in SDMs (Table 2). For an evaluation of these three values we used the approach recommended by *Fielding & Bell (1997)* and *Allouche, Tsoar & Kadmon (2006)*, we found our model usually

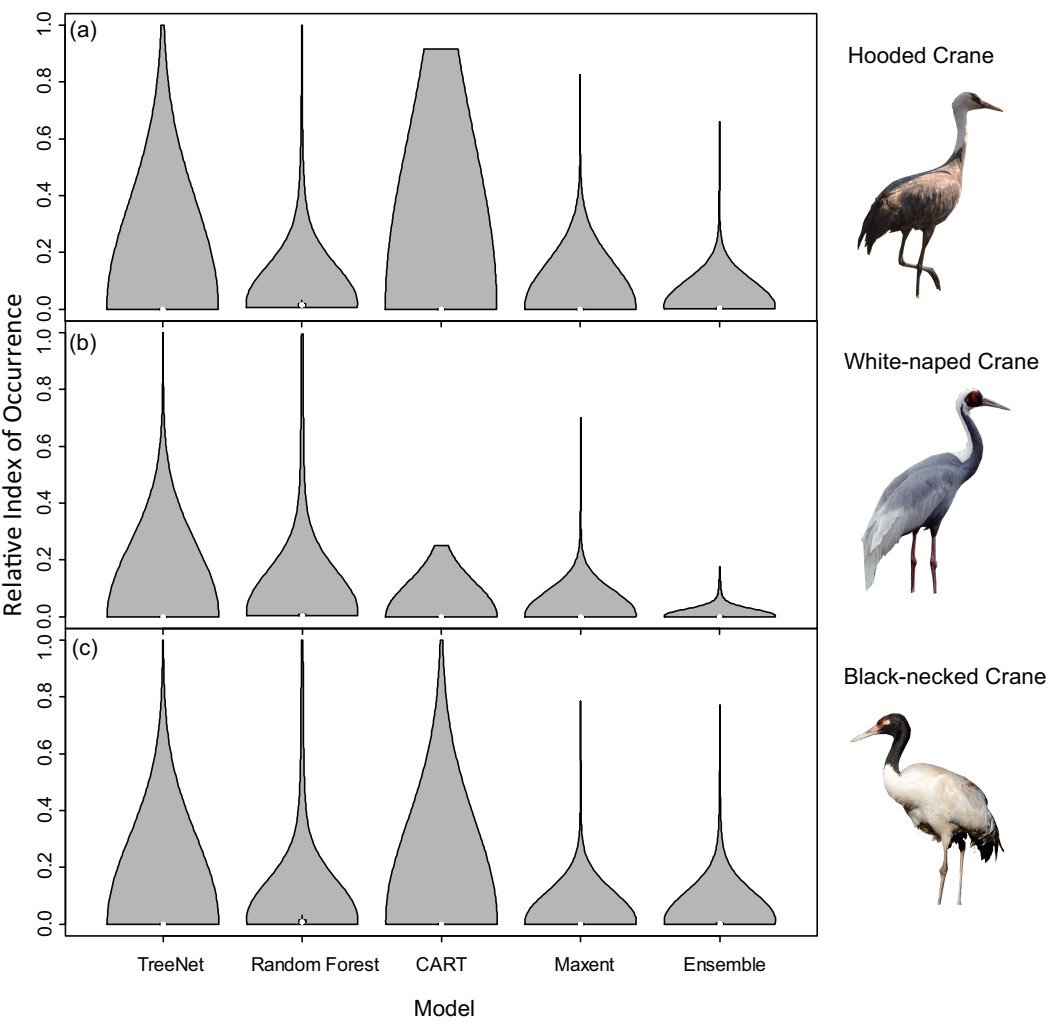

**Figure 8** **Violin plots of Relative Index of Occurrence (RIO) values for four SDMs and Ensemble model for three cranes based on calibration data from Threatened Birds of Asia.** (A) Violin plots for Hooded Cranes, (B) violin plots for White-naped Cranes, (C) violin plots for Black-necked Cranes.

didn't obtain perfect performance, and some of them were 'fair' in their performance. However, for macro-ecology this is more than reasonable and ranks rather high. It's good conservation progress! We identified Random Forest as always the highest performing. These results are consistent with the results of violin plots of the Relative Index of Occurrence (RIO) using tracking as well as literature data (Figs. 3 and 4), and well as matching the spatial assessment results (Figs. 5–7). We recommend when modelers assess model performance they should not only depend alone on some metric (such as AUC and TSS), but also should base their assessments on the combined use of visualization and expert knowledge. That means modelers should also assess how the species distribution map actually looks and how it links with real data (see *Huettmann & Gottschalk, 2011*). Spatial assessment metrics from alternative data should matter the most. Expert experience and ecological common knowledge of the species of interest could sometimes also be highly

effective (*Drew & Perera, 2011*), albeit nonstandard, evaluation methods (see *Kandel et al., 2015* for an example). Additionally, one alternative method for rapid assessment we find is to use a reliable SDM, and thus Random Forest would be a good choice in the future given our consistent results (Figs. 3–7) in this study, which involved three species, a vast landscape to conserve, and only limited data. Our work certainly helps to inform conservation decisions for cranes in Northeast Asia.

### Limitations and future work

Our study is not without limitations: (1) so far, only three species of cranes are used as a test case in our study because nest data for rare species in remote areas are usually sparse, and; (2) all our species study areas are rather vast and confined to East-Asia. For future work, we would apply Random Forest in more species and in more geographical conditions with differently distributed features for a first rapid assessment and baseline to be mandatory for better conservation, e.g., by governments, IUCN and any impact and court decision. We would then apply our prediction results in specifically targeted fieldwork sampling campaigns and assess the model accuracy with field survey results (ground-truthing) and with new satellite tracking and drone data, for instance. This is to be fed directly into the conservation management process.

## ACKNOWLEDGEMENTS

We thank Fengqin Yu, Yanchang Gu, Linxiang Hou, Jianguo Fu, Bin Wang, Jianzhi Li, Lama Tashi Sangpo, Baiyu Lamasery, and Nyainbo Yuze for their hard work in the field. Thanks to all data contributors to the book 'Threatened Birds of Asia.' Thanks to the support of State Forestry Administration and Whitley Fund for Nature (WFN). We also thank Salford Systems Ltd. (Dan Steinberg) for providing the free trial version of their data mining and machine learning software to the conservation research community. This is EWHALE lab publication #177.

### Funding

This Project is fund by the National Natural Science Foundation of China (No. 31570532). The funders had no role in study design, data collection and analysis, decision to publish, or preparation of the manuscript.

### Grant Disclosures

The following grant information was disclosed by the authors:
National Natural Science Foundation of China: No. 31570532.

### Competing Interests

The authors declare there are no competing interests.

## Author Contributions

- Chunrong Mi conceived and designed the experiments, performed the experiments, analyzed the data, contributed reagents/materials/analysis tools, wrote the paper, prepared figures and/or tables, reviewed drafts of the paper.
- Falk Huettmann conceived and designed the experiments, analyzed the data, wrote the paper, reviewed drafts of the paper.
- Yumin Guo, Xuesong Han and Lijia Wen contributed reagents/materials/analysis tools, reviewed drafts of the paper.

## Data Availability

The raw data has been supplied as Supplementary File.

## Supplemental Information

Supplemental information for this article can be found online at http://dx.doi.org/10.7717/peerj.2849#supplemental-information.

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
