# Peer review of "Why choose Random Forest to predict rare species distribution with few samples in large undersampled areas? Three Asian crane species models provide supporting evidence"

_PeerJ, doi:10.7717/peerj.2849_

## Round 0.1 · original submission · Minor Revisions

As you can see, both my reviewers like your paper and so do I. Please fix these minor problems and return the revision as soon as you can.

When PeerJ accepts a paper it gets published "as it is" (i.e. without further copyediting). I recommend that you use a programme such as Grammarly to check your English to ensure that it's the best it can be.

Reviewer 1 ·

Basic reporting

This article is well written and answers an very important question. The structure is clear and figures are relevant.

Experimental design

This study has a very good design. It carried out model evaluation with different measures and with two independent test datasets. So the comparison of models is legitimate. However, there are still some parts that the authors need to address. For the 21 variables that were selected to build models, you need to show if high correlation exists between some variables. The collinearity will influence the validity of your models. For the validation, please report detail number, percentage on how many test data are correctly predicted.

Validity of the findings

The data and methods used in the study are appropriate and valid.

Additional comments

I recommend to accept this paper with minor revisions. This paper answers a good question with a careful design.

Reviewer 2 ·

Basic reporting

The manuscript requires to be looked over and edited throughout for better english.

Experimental design

Line 158: Some additional information on how variables were reduced is necessary. Was there clear evidence that variables were not autocorrelated prior to running the models?

Line 177; also lines 213-214: A check to see if the pseudo-replicated points did not overlap with points obtained with satellite tracking will be important.

Validity of the findings

No comment

Additional comments

I like how the authors have included both the strengths and the shortcomings of their methods ensuring that the findings have high value for conservationists wishing to use the statistical methods described.

---

## Round 0.2 · accepted · Accept

Thank you for attending to these changes quickly.